# Correlates of cervical cancer awareness among women aged 30–49 in five sub-Saharan African nations: Evidence from the Demographic and Health Survey (DHS)—2017–2023

Daniel J. Olivieri[1]*, McKenna C. Eastment[2], Noleb Mugisha[3], Manoj P. Menon[4]

**1** Department of Medicine, Internal Medicine Residency Program, University of Washington, Seattle, Washington, United States of America, **2** Division of Allergy and Infectious Disease, Department of Medicine, Seattle, Washington, United States of America, **3** Uganda Cancer Institute, Kampala, Uganda, **4** Fred Hutchinson Cancer Center, Seattle, Washington, United States of America

\* doliv@uw.edu

## Abstract

Cervical cancer is the leading cause of cancer-related mortality in low- and middle-income countries (LMICs). Prior studies associate high cervical cancer awareness with reductions in cervical cancer incidence. In this study, we utilize nationally representative Demographic and Health Surveys Program (DHS) to analyze correlates of cervical cancer awareness to inform global strategies. All DHS surveys between 2017–2023 were queried for questions on cervical cancer awareness. Socio-demographic variables (e.g., age, marital status), socioeconomic variables (e.g., education, wealth, literacy) and variables pertaining to healthcare decision making, distance traveled, intimate partner violence (IPV), and female genital mutilation/circumcision (FGC/M)) were extracted. Sample weights were applied, and logistic regressions were performed. Variables with p < 0.20 were included in multivariate analysis. Data was obtained from 30,214 women aged 30–49 years old living in Benin, Cameroon, Madagascar, Mauritania, and Mozambique, 19,403 of whom were asked questions on cervical cancer awareness. Cervical cancer awareness varied from 53% in Cameroon to 12% in Benin. Literacy, frequency of watching television, mobile telephone ownership, visiting a local healthcare facility and hormonal contraceptive use were associated with increased cervical cancer awareness, while lack of healthcare decision making independence was associated with decreased awareness after multivariate adjustment. Women who experienced emotional IPV were associated with increased awareness in Cameroon. Less than 4% of all women were screened for cervical cancer. Given the known association between awareness and screening, targeted efforts to increase awareness among women without communication modalities has the potential to reduce global cervical cancer disparities. Potential strategies include co-locating cervical cancer awareness programs with

**Data availability statement:** All data is publicly available third-party data that is not owned or collected by the authors. All data can be accessed upon request from the DHS Survey website online (https://dhsprogram.com/ ) with a brief online application. No authors were granted any special access privileges while obtaining access to this data set. No data was disseminated beyond immediate research team members who obtained DHS approval. Written informed consent was obtained from the parent/guardian of each participant under 18 years of age per ICF International and DHS Protocols.

**Funding:** This work was supported by the UW/Fred Hutch Center for AIDS Research (CFAR) (P30 AI027757 to MM). The funders had no role in study design, data collection and analysis, decision to publish, or preparation of the manuscript.

**Competing interests:** The authors have declared that no competing interests exist.

public health programs and implementing large-scale telecommunication outreach programs to improve awareness.

## Introduction

Cervical cancer is the leading cause of cancer-related death among women in sub-Saharan Africa (SSA) [1–3]. Although the availability of screening and prevention tools (e.g., visual inspection with acetic acid (VIA), human papillomavirus (HPV) testing and vaccinations) has increased, these benefits are not shared amongst all women. The burden of cervical cancer disproportionately affects women living in low- and middle-income countries (LMICs), where nearly 85% of all cervical cancer deaths occur [1,3,4]. Cervical/uterine cancer remains the most common cancer in 48% of SSA nations per World Health Organization (WHO) estimates. Nearly 117,944 women in SSA were diagnosed in 2022, leading to approximately 76,140 deaths (Age Standardized Rate = 22.6, Crude Rate = 13.2) [5,6]. Prior studies attribute global disparities in cervical cancer to differences in human development and gender inequity indices [7,8]. On an individual level, differences in screening rates are a major contributor to this global crisis; less than 15% of all women ages 30–49 in LMICs have received cervical cancer screening, compared to 84% in high-income countries (HICs) [9].

A cornerstone of global cervical cancer prevention is increasing screening accessibility to identify pre-cancerous lesions. Although VIA testing and cytology are more accessible than DNA-based testing, the clinical utility of HPV DNA-based testing is superior. As such, the World Health Organization's (WHO) 2021 Cervical Cancer Screening Guidelines recommend HPV DNA-based testing for women every 5–10 years starting at age 30 as part of their 2020 Global Strategy for Cervical Cancer Elimination [10–12]. Additionally, HPV testing can be self-collected, which has been shown to increase participation among women less likely to be screened [13,14]. Point-of-care or self-collected primary HPV screening can foster a patient-centered approach to reduce mortality, with a number needed to screen (NNS) to prevent one death from cervical cancer of approximately 358 women [15]. This strategy is included in the WHO's 90-70-90 triple-intervention targets, which aim to achieve HPV vaccination rates of 90% or higher, screen >70% of women by age 35 and 45, and ensure >90% of women with pre-cancerous or invasive lesions have access to treatment and care [16]. Provided the triple-intervention targets are met, an estimated 74.1 million cervical cancer cases and 62.6 million deaths, representing a reduction of 97% and 99% respectively, could be prevented by 2120 [17]. Importantly, the 2021 WHO Guidelines note that cervical cancer screening for people living with HIV (PLWH) should be initiated starting at age 25, rather than age 30 for people living without HIV (PLWoH) due to a higher risk of cervical cancer [10].

Cervical cancer awareness — referring to the understanding or knowledge of cervical cancer in any capacity — remains one of the most frequent limiting factors to screening across various settings and countries [18–21] and has been correlated

with socioeconomic status (SES) including wealth, educational attainment, and insurance, as well as sociodemographic factors such as marital status and age [11,20,22–24]. Additionally, one recent study from Jordan demonstrated an association between intimate partner violence (IPV) and cervical cancer screening [25]. Using data from the 2017–2018 Demographic Health Survey (DHS), researchers documented an inverse relationship between experiencing emotional violence and cervical cancer screening awareness (HR = 0.28 [0.07–0.49]), though overall screening rates were unaffected (n = 127/935 screened; OR = 0.28 [0.07–0.49]) [25]. Similarly, female genital mutilation/circumcision (FGM/C), a risk factor for cervical cancer and a form of gender-based violence, represents a potential covariate for cervical cancer awareness, though this has not yet been analyzed in literature to date [26–28].

Although data is limited, awareness appears to vary dramatically across SSA nations, reflecting country-specific norms and challenges to screening. In Kenya, approximately 80% of women queried were aware of cervical cancer whereas in Benin only 10% of women were aware[19,22,29]. Describing correlates in cervical cancer awareness in various SSA nations can inform public health strategies, including the WHO 90-70-90 triple-intervention strategy to reduce the global burden of cervical cancer [1,11,30]; global efforts to reduce the burden cervical cancer should address screening, treatment, access to care, vaccination uptake programs, care follow-up, and cervical cancer among people living with HIV; however, such analysis is beyond the scope of this project. Herein we analyze the role of various sociodemographic and healthcare variables on cervical cancer awareness among the most recently published DHS surveys from 2017–2023 including Benin, Cameroon, Madagascar, Mauritania, and Mozambique. These five countries represent the only SSA nations with DHS questions on cervical cancer awareness published between 2017–2023. Cervical cancer is the most or second most common cancer in these nations, as an estimated 701 Beninese women (ASR = 18.1), 5,525 Cameroonian women (ASR = 33.1), 4,060 Malagasy women (ASR = 41.8), 468 Mauritanian women (ASR = 28.3), and 5,456 Mozambique women (ASR = 47.8) are afflicted with cervical cancer each year [5,6].

## Materials and methods

### Ethics statement

IRB approval for use of these DHS data sets was obtained by ICF International and each respective country-level IRB board. Before each interview was conducted by ICF International, an informed consent statement was read to each respondent, who choose to either accept or decline to participate. Additionally, ICF International's informed consent statement emphasized voluntary participation and confidentiality, and assured respondents may refuse to answer any question or terminate participation at any time. For participants under 18 years of age, a parent or guardian provided informed consent to ICF International prior to participation by a child or adolescent. In summary, this secondary analysis of data utilized publicly available, de-identified third-party data and thus was exempted from standardized IRB review. Data access was obtained from the DHS survey platform after a brief online application. No authors were granted any special access privileges while obtaining access to this data set. No data was disseminated beyond immediate research team members who obtained DHS approval. Data for each survey can be made available through the DHS program via submitting an online request (https://dhsprogram.com/).

### Data source and population

The Demographic and Health Surveys Program (DHS) are a set of nationally representative, standardized household surveys designed and implemented by ICF International every 3–5 years, which collect data on a host of health indicators (https://dhsprogram.com/methodology/survey-Types/dHs.cfm) [31]. First, all DHS surveys between 2017–2023 were queried for questions on cervical cancer awareness (e.g., search terms "cervical" OR "cervical cancer" OR "cancer"). Eleven countries met initial search criteria for further analysis. Each country's DHS survey was queried via the DHS publicly-accessible Survey Question Database (https://questiondatabase.dhsprogram.com/); additionally, each country's

full survey questionnaire was individually reviewed for questions on cervical cancer awareness via publicly available DHS Summary Reports. Six countries were excluded due to lack of specific questions on cervical cancer awareness including: Burkina Faso (2021); Cote d'Ivorie (2021); Gabon (2019–2021); Ghana (2022); Kenya (2022); Rwanda (2019–2020). Five countries met full inclusion criteria: Benin (2017–2018), Cameroon (2018), Madagascar (2021), Mauritania (2019–2021), and Mozambique (2022–2023) (Fig 1).

Women aged 30–49 years old who participated in the DHS from 2017–2022 were included in this analysis, which was modeled off similar studies [11,25,32]. Each DHS survey represents one contiguous entity (i.e., the Benin 2017–2018 survey refers to one entry and was not sub-divided further). Last, since only women were queried in the DHS surveys, we refer to DHS respondents as women, while recognizing that the terminology 'individuals with a cervix' may also be utilized when referencing cervical cancer screening efforts.

### Question/Variable selection

Based on prior DHS studies and cervical cancer screening correlates [11,23,25], twenty-four exposure variables were selected and divided into three sub-categories: sociodemographic factors (i.e., age, marital status, number of children, educational attainment, city population density, and country), female genital mutilation/circumcision (FGM/C)), and IPV-related variables (i.e., physical, emotional, or sexual abuse). Composite variables were considered (i.e., combining television and radio use into a composite media consumption variables) but this was ultimately not implemented since it would lead to a loss of granularity on the impact of selected variables on cervical cancer awareness. IPV variables were only collected and correlated to cervical cancer variables among respondents in Mauritania and Cameroon, and FGM/C was only collected in Mauritania. Awareness of cervical cancer ("Ever heard of cervical cancer?"; Answer: "Yes"/ "No") was the principal outcome assessed, and cervical cancer incidence ("Ever been tested for cervical cancer?") was also collected.

### Statistical analysis

DHS surveys meeting initial search criteria were downloaded and extracted utilizing R software (Vienna, Austria). All data was weighted by the R weight variable v005/1,000,000, consistent with prior DHS surveys and recommendations

**Fig 1. Survey Inclusion Criteria among DHS surveys, 2017-2023.** Five countries met full inclusion criteria, whereas six countries were excluded given lack of specific questions on cervical cancer awareness.

per the DHS manual [31,33]. Counts and proportions for each sub-category were calculated and inferential analysis was performed. Bivariate and multivariate binomial regressions were performed for each country to analyze covariate associations with cervical cancer awareness and screening for each demographic, socioeconomic, and IPV-related variable. Bivariate regression analyses are found in S1 Table. Purposeful selection of all covariates (i.e., covariates with a p-value <0.2 in bivariate analysis) were included in the final regression models [34] Each covariate was tested for collinearity, and we utilized a variance inflation factor (VIF) >5 as a barometer for significant collinearity [35]. Covariates with VIF > 5 were subsequently removed from our final multivariable model. Statistical code was adapted from several open-source R platforms and informative DHS guides produced by USAID [31,33]. In our multivariate models, only one FGC/M and IPV variable were ultimately included given collinearity. Finally, mediation analyses were performed to determine the influence cervical cancer awareness on the relationship between socioeconomic and healthcare access variables on cervical cancer screening. Missing values were treated via listwise deletion (complete case analysis) and data was weighted above to proportionally reflect the sampled population as noted above.

Finally, multivariable analyses for the women aged 15–49 in the selected DHS surveys are attached as S2 Table.

## Results

Thirty thousand and two hundred fourteen women aged 30–49 met study inclusion criteria (6,706 women from Benin, 5,370 from Cameroon, 8,029 from Madagascar, 6,713 from Mauritania, and 3,396 from Mozambique), as summarized in Table 1. Most women were between 30–39 years old [62%; inter-country range 58%–78%], lived in rural areas [59%; 47%–73%], and were married [65%; 33%–81%]. Literacy trends aligned with educational attainment as most respondents were able to read an entire sentence [38%; 16%–55%] and attended at least primary education [35%; 16%–47%]. Only 5% [2%–11%] of all patients were insured across all countries. All wealth index income quintiles were proportionally represented [19%–22%], with upper class being slightly more represented in this cohort [22%; 20–24%]. Finally, healthcare decisions were often made jointly with their partner [47%, 38%–45%] or the partner alone made decisions [31%; 13%–37%], though some woman made decisions independently [22%; 12%–33%]. Among women asked questions about cervical cancer (n = 19,403), cervical cancer awareness was limited [39%; 11%–53%] and screening uptake was poor, as only 456 women across five countries had undergone cervical cancer screening [3%; 1%–6%]. Of those tested, 92% (n = 750) were negative/normal, while 4% were positive (n = 30) as self-reported per respondents.

**Benin.** In Benin, 343 women were aware of cervical cancer (11%), as seen in Table 1. Most woman were between 30–39 (61%), married (71%), illiterate (79%), did not receive formal education (73%), and lacked health insurance (98%). Telephone ownership and radio usage were common (59% and 58%, respectively), while internet usage (5%) was not. Women rarely made healthcare decisions interpedently (16%), and most endorsed having decisions made by their husband/partner (46%).

In Table 2, after multivariate adjustment, higher education (AOR = 1.42 [1.05–1.92]), mobile telephone ownership (AOR = 1.89 [1.25–2.87]), internet use (AOR = 2.15 [1.27–3.63]), and hormonal contraceptive use (AOR = 1.33 [1.12–1.57]) were associated with increased awareness. Urban/rural residence, literacy, health insurance, and wealth were not associated with increased awareness.

**Cameroon.** In Cameroon, 2,857 of women were aware of cervical cancer (53%), as seen in Table 1. Of women aware, most had heard of a test for cervical cancer (64%). Most women were married (60%), able to read an entire sentence (55%), achieved at least a primary education (76%), lacked health insurance (97%), and lived in an urban locale (53%). Most women owned a mobile telephone (71%), but did not frequently listen to the radio (59%) or use the Internet (75%) Husbands/partners were frequently involved in the healthcare decision-making process either with (51%) or without (37%) the interviewed woman. Sexual and physical abuse was uncommon but still present (8% and 12%), whereases emotional abuse was more common (31%).

PLOS Global Public Health

**Table 1. Demographic and descriptive statistics per selected DHS surveys, 2017-2023.**

| | Total (N = 30,214; % of total) | Benin (2017–2018; N = 6,706) | Cameroon (2018; N = 5,370) | Madagascar (2021; N = 8,029) | Mauritania (2019–2021; N = 6,713) | Mozambique (2023; N = 3,396) |
|---|---|---|---|---|---|---|
| **Sociodemographic factors: (% of sample)** | | | | | | |
| Age (base: women 30–49) | | | | | | |
| 30–39 | 18,772 (62%) | 4,100 (61%) | 3,369 (63%) | 4,630 (58%) | 4,036 (60%) | 2,637 (78%) |
| 40–49 | 11,442 (38%) | 2,606 (39%) | 2,001 (37%) | 3,399 (42%) | 2,677 (40%) | 759 (22%) |
| Location of housing | | | | | | |
| Urban | 12,469 (41%) | 2,993 (45%) | 2,826 (53%) | 2,152 (27%) | 3,387 (50%) | 1,111 (33%) |
| Rural | 17,745 (59%) | 3,713 (55%) | 2,544 (47%) | 5,877 (73%) | 3,326 (50%) | 2,285 (67%) |
| Current marital status | | | | | | |
| Never in union | 1,322 (4%) | 103 (2%) | 513 (10%) | 363 (5%) | 272 (4%) | 71 (2%) |
| Married | 19,789 (65%) | 4,735 (71%) | 3,248 (60%) | 5,258 (66%) | 5,422 (81%) | 1,126 (33%) |
| Living with partner | 4,566 (15%) | 1,219 (18%) | 768 (14%) | 884 (11%) | 0 | 1,695 (50%) |
| Widowed | 1,283 (4%) | 297 (4%) | 323 (6%) | 359 (4%) | 190 (3%) | 114 (3%) |
| Divorced | 1,427 (4%) | 114 (2%) | 121 (2%) | 252 (3%) | 829 (12%) | 111 (3%) |
| No longer living together/separated | 1,827 (6%) | 238 (4%) | 397 (7%) | 913 (11%) | 0 | 279 (8%) |
| Literacy | | | | | | |
| Cannot read at all | 15,061 (50%) | 5,285 (79%) | 1,755 (33%) | 2,568 (32%) | 3,324 (50%) | 2,129 (63%) |
| Able to read parts of sent. | 3,563 (12%) | 339 (5%) | 619 (12%) | 1,240 (15%) | 1,065 (16%) | 300 (9%) |
| Able to read whole sent. | 11,448 (38%) | 1,041 (16%) | 2,978 (55%) | 4,203 (52%) | 2,261 (34%) | 965 (28%) |
| No card with req language | 55 (<1%) | 35 (<1%) | 7 (<1%) | 0 | 12 (<1%) | 1 (<1%) |
| Blind/Visually impaired | 87 (<1%) | 6 (<1%) | 11 (<1%) | 18 (<1%) | 51 (<1%) | 1 (<1%) |
| Education level | | | | | | |
| No education | 12,576 (42%) | 4,873 (73%) | 1,273 (24%) | 1,920 (24%) | 3,198 (48%) | 1,312 (39%) |
| Primary | 10,439 (35%) | 1,070 (16%) | 1,858 (35%) | 3,784 (47%) | 2,294 (34%) | 1,433 (42%) |
| Secondary | 6,187 (20%) | 649 (10%) | 1,894 (35%) | 2,027 (25%) | 1,069 (16%) | 548 (16%) |
| Higher | 1,012 (3%) | 114 (2%) | 345 (6%) | 298 (4%) | 152 (2%) | 103 (3%) |
| Presence of health insurance | | | | | | |
| No | 28,786 (95%) | 6,599 (98%) | 5,202 (97%) | 7,684 (96%) | 5,968 (89%) | 3,333 (98%) |
| Yes | 1,428 (5%) | 107 (2%) | 168 (3%) | 345 (4%) | 745 (11%) | 63 (2%) |
| Wealth index for Urban/Rural | | | | | | |
| Lower Class | 5,876 (19%) | 1,307 (19%) | 901 (17%) | 1,611 (20%) | 1,316 (20%) | 741 (22%) |
| Lower Middle Class | 5,694 (19%) | 1,308 (20%) | 946 (18%) | 1,540 (19%) | 1,256 (19%) | 644 (19%) |
| Middle Class | 5,751 (19%) | 1,292 (19%) | 1,083 (20%) | 1,609 (20%) | 1,180 (18%) | 587 (17%) |
| Upper Middle Class | 6,163 (20%) | 1,324 (20%) | 1,177 (22%) | 1,647 (21%) | 1,278 (19%) | 737 (22%) |
| Upper Class | 6,730 (22%) | 1,475 (22%) | 1,263 (24%) | 1,622 (20%) | 1,683 (25%) | 687 (20%) |
| Frequency of listening to the radio | | | | | | |
| Not at all | 14,972 (50%) | 2,806 (42%) | 3,183 (59%) | 3,406 (42%) | 3,397 (51%) | 2,180 (64%) |
| Less than once a week | 7,199 (24%) | 1,417 (21%) | 1,129 (21%) | 1,984 (25%) | 2,121 (32%) | 548 (16%) |
| At least once a week | 8,043 (27%) | 2,483 (37%) | 1,058 (20%) | 2,639 (33%) | 1,195 (18%) | 668 (20%) |
| Frequency of watching television | | | | | | |
| Not at all | 17,948 (59%) | 4,321 (64%) | 2,404 (45%) | 5,566 (69%) | 3,337 (50%) | 2,320 (68%) |
| Less than once a week | 4,402 (15%) | 1,083 (16%) | 649 (12%) | 955 (12%) | 1,448 (22%) | 267 (8%) |
| At least once a week | 7,864 (26%) | 1,302 (19%) | 2,317 (43%) | 1508 (19%) | 1,928 (29%) | 809 (24%) |
| Frequency of reading newspaper or magazine | | | | | | |
| Not at all | 26,375 (87%) | 6,282 (94%) | 4,367 (81%) | 6,768 (84%) | 5,829 (86%) | 3,129 (92%) |

*(Continued)*

| | Total (N = 30,214; % of total) | Benin (2017–2018; N = 6,706) | Cameroon (2018; N = 5,370) | Madagascar (2021; N = 8,029) | Mauritania (2019–2021; N = 6,713) | Mozambique (2023; N = 3,396) |
|---|---|---|---|---|---|---|
| Less than once a week | 2,558 (8%) | 239 (4%) | 659 (12%) | 899 (11%) | 591 (9%) | 170 (5%) |
| At least once a week | 1,281 (4%) | 185 (3%) | 344 (6%) | 362 (5%) | 293 (4%) | 97 (3%) |
| Owns a mobile telephone | | | | | | |
| No | 12,501 (41%) | 2,773 (41%) | 1,555 (29%) | 5,161 (64%) | 1,171 (17%) | 1,841 (54%) |
| Yes | 17,713 (59%) | 3,933 (59%) | 3,815 (71%) | 2,868 (36%) | 5,542 (83%) | 1,555 (46%) |
| Use of internet | | | | | | |
| Never | 25,118 (83%) | 6,330 (94%) | 4,039 (75%) | 7,246 (90%) | 4,703 (70%) | 2,800 (82%) |
| Yes, last 12 months | 4,617 (15%) | 345 (5%) | 1,192 (22%) | 713 (9%) | 1,808 (27%) | 559 (16%) |
| Yes, before last 12 months | 479 (2%) | 31 (<1%) | 139 (3%) | 70 (1%) | 202 (3%) | 37 (1%) |
| Person who usually decides on respondent's health care | | | | | | |
| Respondent alone | 5,365 (22%) | 947 (16%) | 476 (12%) | 2,027 (33%) | 1,058 (20%) | 857 (30%) |
| Respondent and husband/partner | 11,393 (47%) | 2,264 (38%) | 2,061 (51%) | 3,316 (54%) | 2,511 (46%) | 1,241 (44%) |
| Husband/partner alone | 7,507 (31%) | 2,727 (46%) | 1,471 (37%) | 786 (13%) | 1,806 (33%) | 717 (25%) |
| Some else | 56 (<1%) | 10 (<1%) | 4 (<1%) | 6 (<1%) | 33 (<1%) | 3 (<1%) |
| Other | 30 (<1%) | 6 (<1%) | 4 (<1%) | 3 (<1%) | 14 (<1%) | 3 (<1%) |
| Distance to the health facility | | | | | | |
| No problem | 11,239 (37%) | 2,120 (32%) | 2,144 (40%) | 2,760 (34%) | 2,744 (41%) | 1,471 (43%) |
| Big problem | 18,975 (63%) | 4,586 (68%) | 3,226 (60%) | 5,269 (66%) | 3,969 (59%) | 1,925 (57%) |
| Not a big problem | | | | | | |
| **Health behaviors: (% of sample)** | | | | | | |
| Visited health facility in the last 12 months | | | | | | |
| No | 14,927 (52%) | 3,906 (58%) | 2,595 (48%) | 4,487 (56%) | 2,975 (55%) | 964 (28%) |
| Yes | 13,996 (48%) | 2,800 (42%) | 2,775 (52%) | 3,542 (44%) | 2,447 (45%) | 2,432 (72%) |
| Tobacco use | | | | | | |
| Does not smoke | 29,727 (98%) | 6,627 (99%) | 5,342 (99%) | 7,903 (98%) | 6,539 (97%) | 3,316 (98%) |
| Every day | 213 (1%) | 6 (<1%) | 12 (<1%) | 58 (1%) | 125 (2%) | 12 (<1%) |
| Some days | 274 (1%) | 73 (1%) | 16 (<1%) | 68 (1%) | 49 (1%) | 68 (2%) |
| Hormonal contraceptive use | | | | | | |
| No, no use | 14,383 (54%) | 4,182 (62%) | 3,209 (60%) | 2,231 (28%) | 4,761 (71%) | NA |
| Yes, used outside calendar year | 4,165 (16%) | 652 (10%) | 2,161 (40%) | 887 (11%) | 465 (7%) | NA |
| Yes, use in calendar year | 8,270 (31%) | 1,872 (28%) | 0 | 4,911 (61%) | 1,487 (22%) | NA |
| Female circumcision/genital cutting: Ever heard of female circumcision? | | | | | | |
| No | 142 (4%) | NA | NA | NA | 142 (4%) | NA |
| Yes | 3,218 (96%) | NA | NA | NA | 3,218 (96%) | NA |
| Female circumcision/genital cutting: Ever heard of genital cutting (probed)? | | | | | | |
| No | 122 (86%) | NA | NA | NA | 122 (86%) | NA |
| Yes | 20 (14%) | NA | NA | NA | 20 (14%) | NA |
| Respondent underwent female circumcision/ genital mutilation (Base: ever heard of female circumcision) (N = 7495) | | | | | | |
| No | 954 (29%) | NA | NA | NA | 954 (29%) | NA |
| Yes | 2,284 (71%) | NA | NA | NA | 2,284 (71%) | NA |

*(Continued)*

**Table 1.** (Continued)

| | Total (N = 30,214; % of total) | Benin (2017–2018; N = 6,706) | Cameroon (2018; N = 5,370) | Madagascar (2021; N = 8,029) | Mauritania (2019–2021; N = 6,713) | Mozambique (2023; N = 3,396) |
|---|---|---|---|---|---|---|
| **Intimate partner violence-related screening: (% of sample)** | | | | | | |
| Physical abuse: ever been pushed, shook, or had something thrown by husband/partner? | | | | | | |
| Never | 8,966 (89%) | 2,248 (90%) | 2,084 (82%) | 2,782 (90%) | 1,852 (97%) | NA |
| Often | 236 (2%) | 67 (3%) | 112 (4%) | 42 (1%) | 15 (<1%) | NA |
| Sometimes | 378 (4%) | 89 (4%) | 157 (6%) | 117 (4%) | 15 (<1%) | NA |
| Yes, but not last 12 months | 478 (5%) | 85 (3%) | 202 (8%) | 161 (5%) | 30 (3%) | NA |
| Emotional abuse: ever experience any emotional violence? | | | | | | |
| No | 7,015 (70%) | 1,561 (63%) | 1,775 (69%) | 2,062 (66%) | 1,617 (85%) | NA |
| Yes | 3,043 (30%) | 928 (37%) | 780 (31%) | 1,040 (34%) | 295 (15%) | NA |
| Emotional abuse: ever been humiliated by husband/partner? | | | | | | |
| Never | 8,170 (81%) | 1,836 (74%) | 2,054 (80%) | 2,529 (82%) | 1,751 (91%) | NA |
| Often | 454 (5%) | 172 (7%) | 139 (5%) | 94 (3%) | 49 (3%) | NA |
| Sometimes | 789 (8%) | 293 (12%) | 194 (8%) | 231 (7%) | 71 (4%) | NA |
| Yes, but not in last 12 months | 645 (6%) | 188 (8%) | 168 (7%) | 248 (8%) | 41 (2%) | NA |
| Sexual abuse: physically forced to perform sexual acts (N = 3,307) | | | | | | |
| Never | 7,447 (91%) | 2,319 (93%) | 2,320 (91%) | 2,808 (89%) | NA | NA |
| Often | 117 (1%) | 33 (1%) | 49 (2%) | 35 (1%) | NA | NA |
| Sometimes | 315 (4%) | 73 (3%) | 98 (4%) | 144 (5%) | NA | NA |
| Yes, but not last 12 months | 267 (3%) | 64 (3%) | 88 (3%) | 115 (5%) | NA | NA |
| **Screening/awareness questions: (% of sample)** | | | | | | |
| Ever heard of cervical cancer? | | | | | | |
| No | 11,781 (61%) | 2,836 (89%) | 2,513 (47%) | 2,543 (62%) | 1,720 (51%) | 2,169 (64%) |
| Yes | 7,622 (39%) | 343 (11%) | 2,857 (53%) | 1,555 (38%) | 1,640 (49%) | 1,227 (36%) |
| Ever heard of test for cervical cancer? (Base: heard of cervical ca) | | | | | | |
| No | 3,132 (41%) | 162 (47%) | 1,028 (36%) | 697 (45%) | 922 (56%) | 323 (26%) |
| Yes | 4,490 (59%) | 181 (53%) | 1,829 (64%) | 858 (55%) | 718 (44%) | 904 (74%) |
| Ever been tested for cervical cancer? | | | | | | |
| Yes | 456 (3%) | 29 (1%) | 311 (6%) | 87 (2%) | 29 (1%) | NA |
| No | 15,475 (97%) | 3,145 (99%) | 5,049 (94%) | 4,010 (98%) | 3,271 (97%) | NA |
| Unsure | 77 (<1%) | 5 (<1%) | 10 (<1%) | 1 (<1%) | 60 (2%) | NA |
| Last test result for cervical cancer | | | | | | |
| Normal/Negative | 750 (92%) | 26 (81%) | 294 (95%) | 82 (94%) | 26 (90%) | 322 (93%) |
| Abnormal/Positive | 30 (4%) | 2 (6%) | 7 (2%) | 3 (3%) | 2 (7%) | 16 (5%) |
| Suspicion of cancer | 10 (1%) | 0 | 3 (1%) | 0 | 1 (3%) | 6 (2%) |
| Not clear/Undetermined | 17 (2%) | 4 (13%) | 6 (2%) | 1 (1%) | 0 | 6 (2%) |
| Didn't receive results | 2 (<1%) | 0 | 0 | 0 | 0 | 2 (1%) |
| Don't know | 4 (<1%) | 0 | 1 (<1%) | 1 (1%) | 0 | 2 (1%) |

PLOS Global Public Health

**Table 2. Multivariate weighted analysis for cervical cancer awareness among women aged 30–49 for Benin, Cameroon, Madagascar, Mauritania, and Mozambique per Demographic Health Survey 2017–2023.**

| | Benin | | Cameroon | | Madagascar | | Mauritania | | Mozambique | |
|---|---|---|---|---|---|---|---|---|---|---|
| | AOR (95% CI) | *P*-value | AOR (95% CI) | *P*-value | AOR (95% CI) | *P*-value | AOR (95% CI) | *P*-value | AOR (95% CI) | *P*-value |
| Age (base: women 30–49) | NA | NA | 1.04 (1.02–1.07) | P<0.001 | NA | NA | NA | NA | 1.02 (0.99–1.04) | P=0.3 |
| Location of housing | | | | | | | | | | |
| Urban | --- | | --- | | --- | | --- | | --- | |
| Rural | 0.7 (0.45–1.09) | P=0.11 | 0.7 (0.49–1.01) | P=0.06 | 0.47 (0.36–0.62) | P<0.001 | 0.82 (0.53–1.27) | P=0.38 | 0.83 (0.53–1.3) | P=0.42 |
| Current marital status | | | | | | | | | | |
| Never Married | --- | | --- | | --- | | | | | |
| Current/Previously Married | 0.95 (0.65–1.38) | P=0.78 | 1.51 (1.08–2.13) | P=0.02 | 1 (0.94–1.07) | P=0.91 | NA | NA | NA | NA |
| Literacy | | | | | | | | | | |
| Cannot read at all | --- | | --- | | --- | | --- | | --- | |
| Able to read | 1.15 (0.89–1.48) | P=0.28 | 1.42 (1.17–1.71) | P<0.001 | 1.37 (1.2–1.56) | P<0.001 | 1.26 (1.04–1.52) | P=0.02 | 1.25 (1.03–1.51) | P = 0.02 |
| Education level | | | | | | | | | | |
| No education | --- | | --- | | --- | | --- | | --- | |
| Education (any level) | 1.42 (1.05–1.92) | P=0.02 | 1.78 (1.42–2.24) | P<0.001 | 1.64 (1.38–1.95) | P<0.001 | 1.01 (0.83–1.24) | P=0.89 | 1.04 (0.84–1.28) | P=0.71 |
| Presence of health insurance | | | | | | | | | | |
| No | --- | | --- | | --- | | --- | | --- | |
| Yes | 1.36 (0.65–2.84) | P=0.42 | 0.93 (0.43–2) | P=0.86 | 1.08 (0.76–1.55) | P=0.67 | 0.92 (0.53–1.6) | P=0.77 | 0.89 (0.52–1.55) | P=0.69 |
| Wealth index for Urban/Rural | | | | | | | | | | |
| Lower Class | --- | | --- | | --- | | --- | | --- | |
| >Lower Class | 1.08 (0.92–1.27) | P=0.32 | 1.16 (1.03–1.3) | P=0.01 | 1.19 (1.09–1.3) | P<0.001 | 1.1 (0.98–1.23) | P=0.1 | 1.09 (0.97–1.23) | P=0.13 |
| Frequency of listening to the radio | | | | | | | | | | |
| Not at all | --- | | --- | | --- | | --- | | --- | |
| Less than once a week or more | 1.06 (0.89-1.27) | P=0.5 | 1.08 (0.9–1.28) | P=0.4 | 1.47 (1.31-1.65) | P<0.001 | 0.95 (0.79-1.15) | P=0.6 | 0.94 (0.78-1.14) | P = 0.55 |
| Frequency of watching television | | | | | | | | | | |
| Not at all | --- | | --- | | --- | | --- | | --- | |
| Less than once a week or more | 1.19 (0.96-1.46) | P=0.11 | 1.04 (0.85-1.27) | P=0.72 | 1.03 (0.89-1.2) | P=0.67 | 1.46 (1.17-1.83) | P<0.001 | 1.47 (1.17-1.84) | P<0.001 |
| Frequency of reading newspaper or magazine | | | | | | | | | | |
| Not at all | --- | | --- | | --- | | --- | | --- | |
| Less than once a week or more | 0.8 (0.59-1.08) | P=0.14 | 1.22 (0.9-1.65) | P=0.21 | 1.33 (1.08-1.64) | P=0.007 | 0.92 (0.61-1.39) | P=0.7 | 0.92 (0.61-1.4) | P=0.7 |

*(Continued)*

| | Benin | | Cameroon | | Madagascar | | Mauritania | | Mozambique | |
|---|---|---|---|---|---|---|---|---|---|---|
| | AOR (95% CI) | *P*-value | AOR (95% CI) | *P*-value | AOR (95% CI) | *P*-value | AOR (95% CI) | *P*-value | AOR (95% CI) | *P*-value |
| Owns a mobile telephone | | | | | | | | | | |
| No | --- | | --- | | --- | | --- | | --- | |
| Yes | 1.89 (1.25-2.87) | P=0.003 | 1.34 (0.96-1.88) | P=0.08 | 0.95 (0.78-1.17) | P = 0.65 | 2.02 (1.38-2.94) | P < 0.001 | 1.98 (1.37-2.88) | P<0.001 |
| Use of internet | | | | | | | | | | |
| Never | --- | | --- | | --- | | --- | | --- | |
| Yes, has used | 2.15 (1.27-3.63) | P=0.004 | 1.08 (0.82-1.44) | P=0.58 | 1.96 (1.35-2.86) | P<0.001 | 0.86 (0.63-1.18) | P=0.36 | 0.89 (0.65-1.21) | P=0.44 |
| Person who usually decides on respondent's health care | | | | | | | | | | |
| Respondent alone | --- | | --- | | | | --- | | --- | |
| Other | 0.95 (0.84-1.07) | P=0.37 | 0.76 (0.68-0.85) | P<0.001 | NA | NA | 0.86 (0.76-0.96) | P=0.01 | 0.86 (0.77-0.97) | P=0.01 |
| Distance to the health facility | | | | | | | | | | |
| No problem | --- | | --- | | --- | | --- | | --- | |
| Big problem | 0.79 (0.51-1.22) | P=0.28 | 1.12 (0.84-1.5) | P=0.45 | 0.99 (0.81-1.21) | P=0.89 | 0.79 (0.6-1.08) | P=0.14 | 0.79 (0.58-1.08) | P=0.14 |
| Visited health facility in the last 12 months | | | | | | | | | | |
| No | --- | | --- | | --- | | --- | | --- | |
| Yes | 1.01 (0.72-1.39) | P = 0.97 | 1.72 (1.33-2.21) | P < 0.001 | 1.22 (1.01-1.47) | P=0.04 | 1.62 (1.24-2.1) | P<0.001 | 1.64 (1.26-2.12) | P=0.15 |
| Tobacco use | | | | | | | | | | |
| Non-smoker | --- | | --- | | --- | | --- | | --- | |
| Current smoker | 0.78 (0.35-1.73) | P=0.54 | 1.14 (0.23-5.65) | P=0.87 | 1.14 (0.79-1.65) | P=0.49 | 1.59 (0.82-3.08) | P=0.17 | 1.6 (0.84-3.07) | P=0.004 |
| Hormonal contraceptive use | | | | | | | | | | |
| Not used in past | --- | | --- | | --- | | --- | | --- | |
| Yes, used previously | 1.33 (1.12-1.57) | P = 0.001 | 1.37 (1.05-1.79) | P = 0.02 | 1.24 (1.11-1.38) | P < 0.001 | 1.26 (1.07-1.49) | P = 0.005 | 1.28 (1.08-1.51) | P = 0.09 |
| Respondent underwent female circumcision/genital mutilation (Base: ever heard of female circumcision) (N=7495) | | | | | | | | | | |
| No | NA | NA | NA | NA | NA | NA | --- | | --- | |
| Yes | NA | NA | NA | NA | NA | NA | 0.78 (0.58-1.05) | P = 0.1 | 0.77 (0.57-1.04) | P = 0.09 |
| Emotional abuse: ever experience any emotional violence | | | | | | | | | | |

*(Continued)*

**Table 2.** (Continued)

| | Benin | | Cameroon | | Madagascar | | Mauritania | | Mozambique | |
|---|---|---|---|---|---|---|---|---|---|---|
| | AOR (95% CI) | *P*-value | AOR (95% CI) | *P*-value | AOR (95% CI) | *P*-value | AOR (95% CI) | *P*-value | AOR (95% CI) | *P*-value |
| No | NA | NA | --- | | NA | NA | --- | | --- | --- |
| Yes | NA | NA | 1.38 (1.05-1.82) | P = 0.02 | NA | NA | 1.39 (0.94–2.04) | P = 0.1 | 1.4 (0.95–2.06) | P = 0.09 |

Table 2 Multivariate regression analyses for cervical cancer awareness. All covariates with p values <0.20 were included in the final multivariate regression model. All formulas were weighted per v005/1000000. *** signifies covariate is significant at p<0.001; ** signifies covariate is significant at p<0.01,; * signifies covariate is significant at p<0.05. Confidence intervals and p-values calculated separately for each individual covariate. OR=Odds Ratios. AOR=Adjusted Odds Ratios.

After multivariate adjustment, increasing age (AOR=1.04 [1.02–1.07]), literacy (AOR=1.42 [1.17–1.71]), being married (AOR=1.51 [1.08–2.13]), education (AOR=1.78 [1.42–2.24]), higher wealth index (AOR=1.16 [1.03–1.3]), hormonal contraceptive use (AOR=1.37 [1.05–1.79]), and visiting a healthcare facility in the past year (AOR=1.72 [1.33–2.21]) were associated with increased awareness, as documented in Table 2. In addition, women who did not independently make their own medical decisions were associated with decreased awareness (AOR=0.76 [0.68–0.85]). Last, Cameroonian women who experienced emotional violence endorsed greater awareness of cervical cancer (AOR=1.38 [1.05–1.82]).

**Madagascar.** In Madagascar, 1,555 women were aware of cervical cancer (38%), 55% of which heard of a test for cervical cancer, as seen in Table 1. Most women were between 30–39 (58%). The majority lived in rural areas (73%), were married (66%), literate (60%), attended at least primary education (76%), and lacked health insurance (96%). 58% used the radio and 31% endorsed watching television, whereas internet usage was uncommon (10%). Regarding primary healthcare decision makers, women often made decisions jointly with their husband/partner (54%) or alone (33%), followed by husband/partner alone (13%).

Next, in Table 2, after multivariate adjustment, literacy (AOR=1.37 [1.2–1.56]), education (AOR=1.64 [1.38–1.95]), higher wealth index (AOR=1.19 [1.09–1.3]), listening to the radio (AOR=1.47 [1.31–1.65]), reading the newspaper (AOR=1.33 [1.08–1.64]), internet usage (AOR=1.96 [1.32–2.86]), visiting a healthcare facility in the past year (AOR=1.22 [1.01–1.47]), and hormonal contraceptive use (AOR=1.24 [1.11–1.38]) were associated with increasing awareness. Additionally, women living in rural areas were associated with decreased awareness (AOR=0.47 [0.36–0.62]).

**Mauritania.** In Mauritania, 1,640 women were aware of cervical cancer (49%); only 44% of women aware of cervical cancer endorsed knowledge of a test for cervical cancer, as seen in Table 1. Most women were between 30–39 (60%), lived in rural areas (50%), and married (81%). Few women owned health insurance (11%). Nearly half were literate (50%), and 48% did not received primary education. While physical was uncommon (3%), emotional abuse occurred more frequently (15%).

After multivariate adjustment for sociodemographic factors, in Table 2, literacy (AOR=1.26 [1.04–1.52]), greater frequency of watching television (AOR=1.46 [1.17–1.83]), mobile telephone ownership (AOR=2.02 [1.38–2.94]), visiting a healthcare facility in the last year (AOR=1.62 [1.24–2.1]), and hormonal contraceptive use (AOR=1.26 [1.07–1.49]) were associated with increased awareness. Women who did not independently make their own medical decisions were associated with decreased awareness (AOR=0.86 [0.76–0.96]).

**Mozambique.** In Mozambique, 1,227 of the women surveyed were aware of cervical cancer (36%), as seen in Table 1. Of those aware, 74% had heard of a test for cervical cancer. Most women were young, with 78% being between 30–39 years old and 22% between 40–49 years old. Most lived in rural areas (67%), lived with a partner (50%) or were married (33%). Most attended primary schooling (42%), though many women sampled were illiterate (63%). The majority did not

own a telephone (54%), watch television (68%), or use the Internet (82%). Most women made healthcare decisions jointly with their partner (44%), though some women made healthcare decisions independently (30%).

After multivariate adjustment, as seen in Table 2, literacy (AOR = 1.25 [1.03–1.51]), greater frequency of watching television (AOR = 1.47 [1.17–1.84]), and mobile telephone ownership (AOR = 1.98 [1.37–2.88]) were associated with increased awareness. Women who did not independently make their own medical decisions were associated with decreased awareness (AOR = 0.86 [0.77–0.97]).

**Additional analyses.** Next, to determine the influence of cervical cancer awareness on the relationship between socioeconomic status and cervical cancer screening, additional mediation analyses were performed. The mediation analyses revealed a significant indirect effect of education on cervical cancer screening through cervical cancer awareness in Benin (ACME = 0.0059 [0.25–0.01]; p < 0.001). The direct effect of education on cervical cancer screening was not significant in this cohort (ADE = 0.1 [−0.003–0.02]; p = 0.11). The total effect of education on cervical cancer screening was 0.02, 38% of which was mediated through cervical cancer awareness. Similar results were found for the wealth index (ACME = 0.002 [0.001–0]; p < 0.001, ADE = −0.0003 [−0.01–0.01]; p = 0.95) in Benin.

Separate mediation analyses demonstrated a significant indirect effect of education on cervical cancer screening through awareness in Cameroon (ACME = 0.01 [0.006–0.02], p < 0.001; ADE = 0.02 [0.007–0.04], p = 0.002; Total Effect = 0.035, Prop Mediated = 0.4), and Madagascar (ACME = 0.008 [0.004–0.01], p < 0.001; ADE = 0.02 [0.01–0.03], p < 0.001; Total Effect = 0.03, Prop Mediated = 0.25).

## Discussion

Despite advances in screening over the past few decades, cervical cancer remains a major global health crisis [1,17,36]. In this secondary analysis of five DHS surveys between 2017–2023 from Benin, Cameroon, Madagascar, Mauritania, and Mozambique, we document significant inter-country variation in cervical cancer awareness and contrast such findings with other DHS surveys in the Global South [11,30,32].

First, literacy and more frequent visits to healthcare facilities were associated with increased cervical cancer awareness after multivariable adjustment in all countries except for Benin (where literacy and healthcare visits were not associated with cervical cancer awareness) and Mozambique (where healthcare visits were not associated with awareness). In contrast, several other markers of socioeconomic status were not associated with cervical cancer awareness including health insurance (for all nations sampled) and wealth index (for Benin, Mauritania, and Mozambique), contrasting with prior studies [11,20,22,23]. Literacy and healthcare utilization may facilitate increased interaction with cervical cancer awareness campaigns. Interventions to reach patients without high literacy or healthcare utilization may include community-based lay health advisors (LHAs) and group sessions [37–39]. Additionally, such cross-country differences highlight the need for individualized, targeted public health programs to increase cervical cancer awareness. Further country-specific qualitative research from public health experts in each SSA nation is needed to better understand and interpret the associations between socioeconomic status and cervical cancer awareness within each nation.

Next, lack of independent healthcare decision-making autonomy was associated with decreased cervical cancer awareness across Cameroon, Mauritania, and Mozambique. Healthcare decision making autonomy has been documented as a correlate of cervical cancer screening uptake among six DHS surveys from 2011–2018 in SSA, though it has not been documented as a correlate of cervical cancer awareness [40]. Prior studies also document an inverse relationship between gender inequity and cervical cancer incidence and mortality [8]. Importantly, increasing agency in healthcare decision-making increases engagement in the healthcare system, promotes informed decision-making, and may further patient-centered screening campaigns [40–42]. Additionally, women who lack agency in the healthcare decision-making – which may represent a form of reproductive coercion [43] – may delay follow-up and seek treatment at rates below peers [44]. Further qualitative research on the role of healthcare decision-making, with focus on racial/ethnic subpopulations, is crucial to reaching this subset of women [40,45].

Third, given the positive association between mobile telephone ownership and cervical cancer awareness across Benin, Mauritania, and Mozambique, cervical cancer awareness campaigns should include communication via mobile telephone (mHealth) given prior studies show effectiveness in low-resource settings [46]. Given that less than 60% of surveyed women owned a cell phone, these results suggest that increasing cell phone ownership in SSA can help advance cervical cancer awareness campaigns. Similarly, given high rates of radio usage in Mauritania, radio-based telecommunication campaigns can help increase awareness in certain settings. Telecommunication platforms have also been proposed as a potential medium to increase awareness among women who do not independently make their own healthcare decisions [40]. Further research investigating the influence of targeted public health campaigns with LHAs and telecommunication devices is an active area of research [37]. Interstingly, we noted a lack of association between distance to the health facility and cervical cancer awareness in each country, possibly due to dissemination of healthcare materials and outreach via other forms of telecommunication. Further research is needed to better understand this lack of association.

Fourth, women experiencing emotional abuse were paradoxically associated with increased cervical cancer awareness in Cameroon after multivariate adjustment. While emotional IPV has been documented as a correlate of cervical cancer screening, it has not been described as a correlate of cervical cancer awareness [25]. Given noted barriers to screening among women endorsing prior experience of emotional IPV [47], this finding may be explained by more frequent healthcare encounters, questionnaire fidelity, and interviewee response bias [25]. Further research is needed to understand this complex relationship, however, namely the willingness for women to endorse and describe their experience as IPV to a DHS interviewer during large-scale survey data collection [47]. Next, in this analysis, women who underwent FGM/C in Mauritania and Mozambique trended towards a decreased association between cervical cancer awareness after multivariable adjustment; thus, programs targeted at reducing FGM/C, a risk factor for cervical cancer disease progression [27], may consider co-implementation with cervical cancer screening programs. Educating traditional healthcare providers, the main performers of FGM/C in this cohort, on the importance of screening in this population is crucial to increasing awareness and therefore screening. Notably, this finding should be interpreted in a cultural-specific context given the declining prevalence of FGM/C rates in Mauritania, where it has been banned for the past two decades [48,49].

Last, given the association between cervical cancer awareness and screening, increasing awareness is an important first step in reducing the global burden of cervical cancer [11,50]. Further, our mediation analyses demonstrate that awareness mediates the effects of various socioeconomic variables on cervical cancer screening. These results suggest that investments in cervical cancer awareness campaigns will improve screening uptake. Differing rates of cervical cancer awareness between Benin, Cameroon, Madagascar, Mauritania, and Mozambique reflect differences in sociocultural norms, public health infrastructure, and country-level development indices. Awareness was highest in Cameroon and second highest in Mauritania despite the lack of an established national screening program, which may reflect differences in country-level wealth such as the Human Development Index and approach to sub-specialized medical care [8,51,52]. The percentage of women screened for cervical cancer was also highest in Cameroon (6%), though it remains far below WHO targets. Notably, several innovative public health efforts in Benin and Madagascar aim to expand access to visual inspection testing with acetic acid and thermal ablation for treatment in primary health systems [53–56]. Benin's screening efforts, led by the Ministry of Health, aim to build capacity among allied health professionals that may be utilized in other public health programs [53,54]. Additionally, HPV self-testing has been proposed as an ethnographic study in Cameroon as a tool to increase uptake among rural and lower SES cohorts [44]. Implementation of locally driven cervical cancer awareness and screening programs can help address these disparities [57].

Importantly, addressing cervical cancer disparities in SSA requires a multi-faceted approach addressing disease awareness, primary prevention (i.e., HPV-16 and HPV-18 vaccinations), and secondary prevention (i.e., screening and treatment), and oncologic sub-specialty care and follow-up. While beyond the scope of this paper, we believe efforts targeted to increase awareness should be partnered within primary and secondary prevention efforts for maximal efficacy. Combining vertical (i.e., cervical cancer-specific) programs with horizontal public-health programs can facilitate

coordination between private and public actors, and program integration driven by local public health actors is crucial to build infrastructure and human capital [44,53]. Additionally, locally driven qualitative research studies are needed to better understand key barriers to awareness and uptake [44]. Last, efforts at increasing awareness and screening should partner with broader capacity-building efforts [58]. Potential modalities include "sandwich fellowships", post-graduate training programs in HICs bookended with training in home LMIC institution [59]. Special consideration should focus on capacity building in each oncologic-subspeciality including gynecologic oncology, medical oncology, radiation oncology, and pathology; indeed, one prior study estimated there are only eleven pathologists, three medical oncologists and three gynecologic oncologists among the 33 million population of Mozambique [58]. Anecdotally, we report a similar post-graduate capacity-building collaboration between Fred Hutch Cancer Center and the Uganda Cancer Center resulting in training of six Ugandan medical oncologists. Such capacity-building efforts play an essential role in addressing the global burden of cervical cancer.

Strengths of this study include robust, systematic analysis of secondary survey data, the high number of women surveyed for cervical cancer questions across all three countries (n = 19,403), the quantity of questions included in this analysis (n = 14 in this analysis), and generalizability to each nation given the representative sampling. Limitations of this study include lack of generalizability to other SSA nations given documented inter-country variation, recall bias, social desirability bias among respondents [11,60]. Additionally, this study was unable to evaluate the rates of cervical cancer awareness among people living with HIV (PLWH), which is notable given the 2021 WHO Cervical Cancer Screening guidelines recommend initiation of cervical cancer screening at age 25 for PLWH. Next, the cross-sectional nature of the DHS surveys limits any causal inference; reverse causality may also help explain the relationship between healthcare decision-making and cervical cancer awareness. This study was also exploratory in nature and thus it is possible that significant statistical associations may be by chance alone. Our methodology accounted for clustering via complex survey design but still may under-estimate variations due to clustering. Of note, we opted to not include additional pooled multivariable analyses in this project given the significant heterogeneity across our five selected countries, variation in year of survey, inability to pool certain variables (i.e., wealth index) across countries, unclear reference group, and recommendations from prior DHS staff as described in several online forums and publications. Further ethnographic research is needed to elucidate cultural-specific and racial/ethnic disparities in cervical cancer awareness. Last, inclusion of FGM/C questions in future DHS surveys may help better elucidate the association between FGM/C awareness and cervical cancer awareness. Finally, we would like to acknowledge individuals working with the Demographic Health Surveys Program (DHS), a publicly accessible data set, for without this research would not be possible.

## Conclusion

Cervical cancer awareness varies significant by country and region in low-resource settings. Significant disparities exist between and within various sub-Saharan African nations, thus highlighting the need for country-specific, locally driven strategies to increase cervical cancer awareness. In this secondary analysis of five DHS surveys, increased literacy, frequency of watching television, mobile telephone ownership, visiting a local healthcare facility and hormonal contraceptive use were associated with increased cervical cancer awareness, while lack of individualized healthcare decision-making autonomy was associated with decreased awareness after multivariate adjustment. In addition, women experiencing emotional abuse was paradoxically associated with decreased awareness in various settings in SSA. In contrast to studies on cervical cancer screening correlates, several socioeconomic factors, including wealth index and health insurance, were not associated with cervical cancer awareness in our multivariate analyses, though caution should be used when interpreting this data.

Given the known association between awareness and screening, targeted efforts to increase awareness among women without communication modalities has the potential to increase global cervical cancer awareness. Potential modalities include co-locating cervical cancer screening with public health programs and implementing large-scale telecommunication outreach programs to improve awareness.

## Supporting information

**S1 Table. Bivariate analyses for cervical cancer awareness for Benin, Cameroon, Madagascar, Mauritania, and Mozambique per Demographic Health Survey 2017–2023, women aged 30–49.** All covariates with p values <0.20 were included in the final multivariate regression model. All formulas were weighted per v005/1000000. *** signifies covariate is significant at p<0.001; ** signifies covariate is significant at p<0.01; * signifies covariate is significant at p<0.05. Confidence intervals and p-values calculated separately for each individual covariate. OR=Odds Ratios. AOR=Adjusted Odds Ratios. NA=Not applicable.
(DOCX)

**S2 Table. Multivariate weighted analysis for cervical cancer awareness for Benin, Cameroon, Madagascar, Mauritania, and Mozambique per Demographic Health Survey 2017–2023, women ages 15–49.** All covariates with p values <0.20 were included in the final multivariate regression model. All formulas were weighted per v005/1000000. *** signifies covariate is significant at p<0.001; ** signifies covariate is significant at p<0.01; * signifies covariate is significant at p<0.05. Confidence intervals and p-values calculated separately for each individual covariate. OR=Odds Ratios. AOR=Adjusted Odds Ratios. NA=Not applicable.
(DOCX)

## Acknowledgments

We would like to acknowledge individuals working with the Demographic Health Surveys Program (DHS), a publicly accessible data set, for without this research would not be possible.

## Author contributions

**Conceptualization:** Daniel Olivieri, McKenna C Eastment, Noleb Mugisha, Manoj P Menon.

**Data curation:** Daniel Olivieri.

**Formal analysis:** Daniel Olivieri.

**Investigation:** Daniel Olivieri.

**Methodology:** Daniel Olivieri, McKenna C Eastment, Manoj P Menon.

**Software:** Daniel Olivieri.

**Supervision:** Manoj P Menon.

**Validation:** Manoj P Menon.

**Visualization:** Daniel Olivieri.

**Writing – original draft:** Daniel Olivieri, Manoj P Menon.

**Writing – review & editing:** Daniel Olivieri, McKenna C Eastment, Noleb Mugisha, Manoj P Menon.

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
