## [Decision Letter · Decision Letter 0]

7 May 2024

PGPH-D-24-00682

Correlates of cervical cancer awareness in Mauritania, Madagascar, and Benin from the Demographic and Health Survey (DHS)—2017-2022

Dear Dr.Olivieri,

Thank you for submitting your manuscript to PLOS Global Public Health. After careful consideration, we feel that it has merit but does not fully meet PLOS Global Public Health’s publication criteria as it currently stands. Therefore, we invite you to submit a revised version of the manuscript that addresses the points raised during the review process.

EDITOR

Please enough reviewer 2. Just address the comments from reviewer 1 and 3 

We look forward to receiving your revised manuscript.

Kind regards,

Peter Bai James, PhD

Academic Editor

Journal Requirements:

Additional Editor Comments (if provided):

Reviewers' comments:

Reviewer's Responses to Questions

**Comments to the Author**

1. Does this manuscript meet PLOS Global Public Health’s publication criteria ? Is the manuscript technically sound, and do the data support the conclusions? The manuscript must describe methodologically and ethically rigorous research with conclusions that are appropriately drawn based on the data presented.

Reviewer #1: Yes

Reviewer #2: No

Reviewer #3: Yes

2. Has the statistical analysis been performed appropriately and rigorously?

Reviewer #1: Yes

Reviewer #2: No

Reviewer #3: Yes

3. Have the authors made all data underlying the findings in their manuscript fully available (please refer to the Data Availability Statement at the start of the manuscript PDF file)?

Reviewer #1: Yes

Reviewer #2: No

Reviewer #3: Yes

4. Is the manuscript presented in an intelligible fashion and written in standard English?

Reviewer #1: Yes

Reviewer #2: No

Reviewer #3: Yes

5. Review Comments to the Author

Reviewer #1: Thank you for the opportunity to review this paper. The authors have done well to explore the correlates of cervical cancer awareness in SSA. While this is a great paper, there are still some pertinent issues that must be addressed.

Abstract

1. The conclusion section of the abstract is more or less like a repetition of the results. It will be great to see very practical policy and practice recommendations there.

Introduction

2. In the first paragraph, the authors present the epidemiology of cervical cancer in LMICs. I advise that the authors include some current statistics regarding the incidence, prevalence, and mortalities of cervical cancer in SSA and in the selected countries.

3. From line 108, the authors introduce the concept of cervical cancer awareness. It will be of importance to let your readers know exactly what you mean by awareness. It is simply having heard of the disease or it goes beyond that?

4. "Categorizing, describing, and elucidating trends in cervical cancer awareness in various SSA ..." Is that the aim of the study? To examine trends? If that is not the case then please revise the sentence.

5. A major concern in this study is the rationale for selecting the three countries. The authors fail to let us appreciate the differences in cervical cancer across the three countries. Moreover, there are more countries with recent DHS data that have cervical cancer as a variable. Why were they not included?

Methods

6. The authors must include more explanatory variables. Variables such as risky sexual behaviors, having some chronic diseases, having visited the healthcare facility, autonomy in healthcare decision-making are available in the DHS and have been found to influence awareness levels. As such, the authors must run the analysis again taking into consideration all of these possible correlates.

7. Why do you have separated analysis for each country? Were the data not appended? The study aims to examine the correlates in SSA. This means that the data must be appended and the country and survey years accounted for in the model.

Results

8. The tables presented show the bivariable/bivariate. Where is the multivariable/multivariate results?

9. I advise that the authors append the data and then use a multilevel logistic regression analysis.

10. With the critical analytical issues raised, I cannot review the discussion now. As running the analysis again is likely to change the results.

Reviewer #2: I decline to review.I decline to review.I decline to review.I decline to review.I decline to review.I decline to review.I decline to review.I decline to review.I decline to review.I decline to review.I decline to review.I decline to review.I decline to review.I decline to review.I decline to review.I decline to review.I decline to review.I decline to review.I decline to review.I decline to review.I decline to review.I decline to review.I decline to review.I decline to review.I decline to review.I decline to review.I decline to review.I decline to review.I decline to review.I decline to review.I decline to review.I decline to review.I decline to review.I decline to review.I decline to review.I decline to review.I decline to review.I decline to review.I decline to review.I decline to review.I decline to review.I decline to review.I decline to review.I decline to review.I decline to review.I decline to review.

Reviewer #3: Article Review and Remarks:

Manuscript Number: PGPH-D-24-00682

Topic: Correlates of Cervical Cancer Awareness in Mauritania, Madagascar, and Benin from the Demographic and Health Survey (DHS) — 2017-2022

Introduction:

The introduction needs to provide more comprehensive information about cervical cancer in Africa. Notably, 18 out of the 20 countries with the highest burden are in Africa (WHO). This underscores the significance of addressing the disease in this region. In addition of the follwing challenges in Africa:

• Vaccination: Weak vaccination programs lead to low coverage and effectiveness.

• Interplay with HIV: High HIV prevalence in Africa exacerbates cervical cancer risks.

• Awareness: General awareness of cervical cancer is very poor.

• Screening: Limited access to screening leads to late diagnosis.

• Access to Care: Difficulties in accessing care due to cost and logistical issues.

Methodology:

Clarify if data from two different surveys were combined in Benin and, if so, explain the rationale behind it.

Address concerns regarding collinearity, as the data may represent the same population.

Findings:

The presentation of health insurance data in Benin is unclear. Simplify the table for better clarity, ensuring percentages add up correctly.

Table 2A should focus on identifying factors associated with unawareness of cervical cancer, as this is the central issue being addressed.

Conclusion:

Overall, the paper is well-constructed but requires improvements in data presentation and analysis to provide clearer insights into the correlates of cervical cancer awareness in these countries.

6. PLOS authors have the option to publish the peer review history of their article (what does this mean? ). If published, this will include your full peer review and any attached files.

**Do you want your identity to be public for this peer review?** For information about this choice, including consent withdrawal, please see our Privacy Policy .

Reviewer #1: No

Reviewer #2: No

Reviewer #3: **Yes: ** Dr Kouamivi Agboyibor, MD, PhDc, MSc, MPH

---

## [Decision Letter · Decision Letter 1]

22 Nov 2024

PGPH-D-24-00682R1

Correlates of cervical cancer awareness among five sub-Saharan African nations from the Demographic and Health Survey (DHS)—2017-2023

Dear Dr. Daniel Olivieri,

Thank you for submitting your manuscript to PLOS Global Public Health. After careful consideration, we feel that it has merit but does not fully meet PLOS Global Public Health’s publication criteria as it currently stands. Therefore, we invite you to submit a revised version of the manuscript that addresses the points raised during the review process.

EDITOR's Comments : Please insert comments here and delete this placeholder text when finished. Be sure to:

I advise that you address  reviewers 1& 2 concerns  with regards to what new information is this adding to the current literature, why the five countries and the issue around the age specification for HPV screening

Please submit your revised manuscript by  the 21st December 2024 . If you will need more time than this to complete your revisions, please reply to this message or contact the journal office at globalpubhealth@plos.org. Please include the following items when submitting your revised manuscript:

We look forward to receiving your revised manuscript.

Kind regards,

Peter Bai James, PhD

Academic Editor

Journal Requirements:

Additional Editor Comments (if provided):

Reviewers' comments:

Reviewer's Responses to Questions

**Comments to the Author**

1. If the authors have adequately addressed your comments raised in a previous round of review and you feel that this manuscript is now acceptable for publication, you may indicate that here to bypass the “Comments to the Author” section, enter your conflict of interest statement in the “Confidential to Editor” section, and submit your "Accept" recommendation.

Reviewer #1: (No Response)

Reviewer #4: (No Response)

Reviewer #5: All comments have been addressed

2. Does this manuscript meet PLOS Global Public Health’s publication criteria ? Is the manuscript technically sound, and do the data support the conclusions? The manuscript must describe methodologically and ethically rigorous research with conclusions that are appropriately drawn based on the data presented.

Reviewer #1: Partly

Reviewer #4: No

Reviewer #5: Yes

3. Has the statistical analysis been performed appropriately and rigorously?

Reviewer #1: No

Reviewer #4: No

Reviewer #5: Yes

4. Have the authors made all data underlying the findings in their manuscript fully available (please refer to the Data Availability Statement at the start of the manuscript PDF file)?

Reviewer #1: Yes

Reviewer #4: Yes

Reviewer #5: Yes

5. Is the manuscript presented in an intelligible fashion and written in standard English?

Reviewer #1: Yes

Reviewer #4: Yes

Reviewer #5: Yes

6. Review Comments to the Author

Reviewer #1: Thank you for the opportunity to reassess the manuscript. I do not know if the wrong file was uploaded. The revised file I assessed has no track changes to indicate changes made. Besides that, I realised that the data was still treated independently rather than appending it.

My basis for rejecting the manuscript at this point stems from the following:

1. There is no novelty in the paper. It adds very little to what we already know about cervical cancer. All of your findings have already been documented in many studies conducted in SSA. (see: https://www.sciencedirect.com/science/article/pii/S1877782121000473;
https://www.ajol.info/index.php/pamj/article/view/212679;
https://bmjopen.bmj.com/content/12/7/e058026.abstract). So, what new thing is this study adding? I would advise the authors to take time to review literature and identify where there are inconsistencies and capitalize on that. You can even decide to investigate some mediation or moderation effects. But presenting just the correlates adds nothing.

2. Why the three selected countries? There are many other countries with DHS (more current) data on cervical cancer screening. So, it beats my imagination why the authors limit themselves to these three if the aim is to investigate the case in SSA.

Reviewer #4: Before I thoroughly review the manuscript on the correlates of cervical cancer awareness among five sub-Saharan African nations using data from the Demographic and Health Survey (DHS) from 2017 to 2023, I identified a significant issue related to the age distribution. The authors do not appear to acknowledge the established guidelines for cervical cancer screening, which recommend that screening should begin at age 25 for women living with HIV and at age 30 for those who are HIV-negative. The study includes girls as young as 15, which makes the findings not scientifically sound. Additionally, the upper age limit of 49 years does not align with the recommendations, although this is understandable given that the DHS collects data for women aged 15 to 49. It is crucial for the authors to address this issue, as it has significant implications for overall screening uptake and the regional findings. I believe this oversight could have been mitigated if the authors had reviewed the WHO 2021 guidelines on screening in the background section. I look forward to your explanation regarding this matter. I recommend the analysis be carried out among women aged 30-49 years.

Reviewer #5: Thank you for the manuscript. I have no further comments.

7. PLOS authors have the option to publish the peer review history of their article (what does this mean? ). If published, this will include your full peer review and any attached files.

**Do you want your identity to be public for this peer review?** For information about this choice, including consent withdrawal, please see our Privacy Policy .

Reviewer #1: No

Reviewer #4: No

Reviewer #5: **Yes: ** Mohamed Boie Jalloh

---

## [Decision Letter · Decision Letter 2]

16 Feb 2025

<h3>PGPH-D-24-00682R2

Correlates of cervical cancer awareness among five sub-Saharan African nations from the Demographic and Health Survey (DHS)—2017-2023

Dear Dr. Daniel Olivieri,

Thank you for submitting your manuscript to PLOS Global Public Health. After careful consideration, we feel that it has merit but does not fully meet PLOS Global Public Health’s publication criteria as it currently stands. Therefore, we invite you to submit a revised version of the manuscript that addresses the points raised during the review process.

EDITOR: Please address reviewer 1 comments as it relates to  need to do multi-level regression analysis given that DHS data is hierarchical in nature . Please provide a tangible reason if you think multi-level regression analysis is not necessary in this case.</h3>

We look forward to receiving your revised manuscript.

Kind regards,

Peter Bai James, PhD

Academic Editor

Journal Requirements:

Additional Editor Comments (if provided):

Reviewers' comments:

Reviewer's Responses to Questions

**Comments to the Author**

1. If the authors have adequately addressed your comments raised in a previous round of review and you feel that this manuscript is now acceptable for publication, you may indicate that here to bypass the “Comments to the Author” section, enter your conflict of interest statement in the “Confidential to Editor” section, and submit your "Accept" recommendation.

Reviewer #1: All comments have been addressed

Reviewer #4: All comments have been addressed

2. Does this manuscript meet PLOS Global Public Health’s publication criteria ? Is the manuscript technically sound, and do the data support the conclusions? The manuscript must describe methodologically and ethically rigorous research with conclusions that are appropriately drawn based on the data presented.

Reviewer #1: Yes

Reviewer #4: Yes

3. Has the statistical analysis been performed appropriately and rigorously?

Reviewer #1: Yes

Reviewer #4: Yes

4. Have the authors made all data underlying the findings in their manuscript fully available (please refer to the Data Availability Statement at the start of the manuscript PDF file)?

Reviewer #1: Yes

Reviewer #4: Yes

5. Is the manuscript presented in an intelligible fashion and written in standard English?

Reviewer #1: Yes

Reviewer #4: Yes

6. Review Comments to the Author

Reviewer #1: Thank you for the opportunity to review the revised manuscript. The authors have addressed the initial comments satisfactorily. While there are studies on the topic from individual SSA countries, none have looked at the regional picture. So, this is a good addition to knowledge.

Minor comment:

1. Revise the title to read as: "Correlates of cervical cancer awareness among women aged 30-39 years in five sub-Saharan African nations: Evidence from the Demographic and Health Survey"

2. I believe that media exposure should be a composite variable. However, if the authors want to keep what they have, then they must provide justifications in the methods section.

3. The DHS is a hierarchical data. As such, it is important to account for the variations due to clustering. This means a multi-level regression analysis should have been done. Please revise the analysis or provide justification for the approach used, especially when you are working with multiple countries.

4. The authors stated: "All models were tested for collinearity with variance inflation factor (VIF) >5 as a barometer for significant collinearity". What was the mean VIF and how did you deal with individual items that had a VIF >5?

5. The authors must document how missing values were handled.

Reviewer #4: My comments have been addressed.

7. PLOS authors have the option to publish the peer review history of their article (what does this mean? ). If published, this will include your full peer review and any attached files.

**Do you want your identity to be public for this peer review?** For information about this choice, including consent withdrawal, please see our Privacy Policy .

Reviewer #1: No

Reviewer #4: No

---

## [Editor Report · Decision Letter 3]

6 Mar 2025

Correlates of cervical cancer awareness among women aged 30-49 in five sub-Saharan African nations: Evidence from the Demographic and Health Survey (DHS)—2017-2023

PGPH-D-24-00682R3

Dear Daniel Olivieri,

We are pleased to inform you that your manuscript 'Correlates of cervical cancer awareness among women aged 30-49 in five sub-Saharan African nations: Evidence from the Demographic and Health Survey (DHS)—2017-2023' has been provisionally accepted for publication in PLOS Global Public Health.

Best regards,

Peter Bai James, PhD

Academic Editor